# Field- and laboratory-based studies on correlates of *Chlamydia trachomatis* transmission by *Musca sorbens*: Determinants of fly-eye contact and investigations into fly carriage of elementary bodies

Ailie Robinson[1]*, Bart Versteeg[2,3], Oumer Shafi Abdurahman[2,4], Innes Clatworthy[5], Gemeda Shuka[4], Dereje Debela[4], Gebreyes Hordofa[4], Laura Reis de Oliveira Gomes[1], Muluadam Abraham Aga[4], Gebeyehu Dumessa[4], Virginia Sarah[6], David Macleod[2,7], Anna Last[2], Matthew J. Burton[2,8], James G. Logan[1,9]

1 Department of Disease Control, LSHTM, London, United Kingdom, 2 International Centre for Eye Health, Clinical Research Department, London School of Hygiene & Tropical Medicine (LSHTM), London, United Kingdom, 3 Knowledge Institute of the Dutch Association of Medical Specialists, Utrecht, The Netherlands, 4 The Fred Hollows Foundation Ethiopia, Addis Ababa, Ethiopia, 5 Natural History Museum, London, United Kingdom, 6 The Fred Hollows Foundation, London, United Kingdom, 7 MRC International Statistics and Epidemiology Group, LSHTM, London, United Kingdom, 8 National Institute for Health Research Biomedical Research Centre for Ophthalmology at Moorfields Eye Hospital NHS Foundation Trust and UCL Institute of Ophthalmology, London, United Kingdom, 9 Arctech Innovation Ltd, Dagenham, United Kingdom

* ailie.robinson@ukhsa.gov.uk

## Abstract

*Musca sorbens* (Diptera: Muscidae) flies are thought to be vectors of the blinding eye disease trachoma, carrying the bacterium *Chlamydia trachomatis* (Ct) between the eyes of individuals. While their role as vectors has been convincingly demonstrated via randomised controlled trials in The Gambia, studies of fly-borne trachoma transmission remain scant and as such our understanding of their ability to transmit Ct remains poor. We examined fly-eye contact and caught eye-seeking flies from 494 individuals (79% aged ≤9 years) in Oromia, Ethiopia. Ct-carrying flies (harbouring Ct DNA) were found to cluster spatially in and nearby to households in which at least one resident had Ct infection. Fly-eye contact was positively associated with the presence of trachoma (disease), lower human body weight and increased human body temperature. Studies of laboratory-reared *M. sorbens* indicated that Ct is found both externally and internally following feeds on Ct culture, with scanning electron microscopy revealing how Ct bodies can cling to fly hairs (setae). Testing for Ct on field-caught *M. sorbens* found fly 'bodies' (thorax, wings and abdomen) to consistently test positive for Ct while legs and heads were infrequently Ct-positive. These studies strongly support the role of *M. sorbens* as vectors of trachoma and highlight the need for improved understanding of fly-borne trachoma transmission dynamics and vector competence.

**Data Availability Statement:** De-identified individual participant data and other datasets on which statistical analysis including summary figures and tables are based, and an accompanying data dictionary defining each field, are available at LSHTM Data Compass (https://doi.org/10.17037/DATA.00003114).

**Funding:** This work was supported by The Wellcome Trust and The Fred Hollows Foundation through the Stronger SAFE Collaborative Award to MJB, VS and JGL (206275/Z/17/Z). The Fred Hollows Foundation applied untied funds (ET18-14-1102) to this project and staff participated in the study design; in the collection, analysis, and interpretation of data; in the writing of the report; and in the decision to submit the paper for publication. The Wellcome Trust had no involvement in the study design; in the collection, analysis, and interpretation of data; in the writing of the report; and in the decision to submit the paper for publication.

**Competing interests:** The authors have declared that no competing interests exist.

## Author summary

Trachoma is a blinding eye disease caused by infection of the eye by the bacterium *Chlamydia trachomatis (Ct)*. Trachoma is responsible for vision impairment or blindness in 1.8 million people and is the leading infectious cause of blindness worldwide. The presence of flies around the eyes of young children is a common feature of life in trachoma endemic communities worldwide. This fly is most often *Musca sorbens*, an understudied species thought to transmit Ct.

In order to understand more about *Musca sorbens* flies and their relationship with *Ct*, the researchers studied 247 households in the Oromia region of Ethiopia, where there is known to be a lot of trachoma. The authors found evidence of *Ct* on flies and that flies in households containing children with *Ct* infection were much more likely themselves to be carrying *Ct*, adding to evidence that *M. sorbens* flies contribute to transmission of the bacterium. They also found that children who weigh less and currently have trachoma were more likely to have flies land on their face. Should further studies clearly elucidate a role for *M. sorbens* as vectors of trachoma, controlling fly populations or preventing fly-eye contact may become an important aspect of trachoma disease control.

## Introduction

Trachoma is a blinding eye disease caused by infection of conjunctival cells by the intracellular bacterium *Chlamydia trachomatis* (Ct). Trachoma is responsible for vision impairment or blindness in 1.8 million people and is the leading infectious cause of blindness worldwide [1–3]. It is a neglected tropical disease and the infection can be treated with antibiotics. Trachoma is transmitted through contact with ocular and nasal secretions from an infected person containing extracellular forms of the bacterium (elementary bodies, EB). This can occur through various routes, including close personal contact, fomites, and eye-seeking flies [4].

*Musca sorbens* is highly attracted to ocular and nasal discharge as a food source. Across Africa, eye-seeking is relatively specific to *M. sorbens*, and although *M. domestica* can also exhibit this behaviour, the species ratio of flies found at the eye is always predominantly *M. sorbens* [5–7]. Eye-seeking flies particularly attack young children [8] who are also the reservoir for Ct infection [6]. *Musca sorbens* is thought to be a mechanical vector, transiently carrying Ct on or in its body [8] and transmitting infection when eye-seeking. As early as the 9th century AD, 'ophthalmia', inflammation of the eye, was suggested to be a consequence of 'the bite of a fly or bug'. [8] Since the late 1880s cases of eye disease, particularly conjunctivitis or trachoma, were observed to increase at times of high fly density [8], and in the mid 20th century community-wide fly reduction was associated with decreased trachoma [9], conjunctivitis [9,10] and ophthalmia [11] rates. These correlations led early investigators to hypothesise that eye-seeking flies were transmitting eye diseases. Recently, controlled clinical trials have investigated the impact of fly control on trachoma rates. One study found fly control alone to reduce trachoma prevalence [5], while another study of fly control combined with antibiotic treatment showed that fly control reduced trachoma prevalence at six months but not one year [12].

The importance of fly-borne transmission in trachoma epidemiology is recognised, with current trachoma control recommendations including environmental improvement to control flies, however, these are rarely implemented. A critical issue is that the importance of flies is not yet accepted by all trachoma control policy makers because of gaps in our knowledge about fly-borne trachoma. More studies are needed to clarify the role of eye-seeking flies in trachoma transmission.

Although they are probably mechanical vectors, the vector competence of *M. sorbens*—the ability of this vector to acquire and subsequently transmit Ct [13]–remains unstudied. It is often speculated that *M. sorbens* carries Ct on its feet and mouthparts [4,5,14]. However, this is possibly an oversimplification given we know many mechanical vectors carry the greatest loads of pathogenic bacterium internally [8], and may subsequently transmit via vomit or faeces. As yet, we do not understand the sequence of events of 'infection', maintenance and transmission of Ct by *M. sorbens*.

We describe field and preliminary laboratory studies that investigate aspects of *M. sorbens*-Ct transmission dynamics. We previously reported an exploration of Ct transmission routes, including flies, in 13 (Ct) ocular-positive and 15 ocular-negative households studied during a population-based survey of 247 households [7]. Here, we extended that study to examine flies from all households in the population survey (within a 5 km$^2$ area). We aimed to better understand fly-borne trachoma transmission dynamics at this study site by mapping the proximity of Ct-positive flies to ocular positive households and exploring associations between fly-eye contact and disease, person, and environmental variables. We then conducted preliminary investigations into the carriage of Ct by flies, by testing separate field-caught fly body parts, and corroborating these results in the laboratory using Ct-fed colonised *M. sorbens*. We further explored external carriage of Ct by *M. sorbens* using scanning electron microscopy.

## Methods

### Ethics statement

The field studies were conducted in accordance with the declaration of Helsinki, and other aspects of these studies have previously been reported [7]. Ethical approval was given by the Ethics Committees of the London School of Hygiene & Tropical Medicine (Reference 17494), Ethiopian Federal Ministry of Science and Technology (MOSHE//RD/141/8082/19) and Oromia Regional Health Bureau (BEFO/DDFDHU/1-89/3515). Verbal consent was obtained from community leaders. Written informed consent was provided by all participants or (for children) their guardians.

### Clinical assessment and Ct testing

All individuals were examined for clinical signs of trachoma and had a conjunctival swab sample taken as described previously [7]. In brief, the upper tarsal conjunctiva of both eyes was examined and graded by a qualified trachoma grader using the simplified Trachoma Grading System [15]. A conjunctival swab sample (ocular swab) was then collected from the left upper tarsal conjunctiva. Swabs were stored on ice packs in the field, transferred for temporary storage in a -20°C freezer before transfer to a -80°C freezer, where they were stored until testing for Ct.

### Entomological field studies

**Study location and design.**    A population-based survey of 247 households was conducted in Shashemane district, Oromia Region, Ethiopia, between April and June 2018 [7]. The survey aimed to include all households in a geographically contiguous area in which at least one child, aged 1–9 years, was resident on the day of enumeration. Full details of the survey are published elsewhere [7].

**Fly-eye observation.**    Per household, fly contacts on two children aged 2–9 years were observed for ten minutes. If children were unavailable or there were not two children aged 2–9 years, children aged >9 years were observed (with a preference for the youngest), if no

children were present adults were observed. All residents of the household underwent a clinical assessment by a qualified trachoma grader as described previously [7]. Person and environmental data were collected for fly-eye individuals including body weight, tympanic temperature, ocular and nasal discharge (as previously defined [7]), ambient temperature, light intensity, relative humidity and the presence of rain or wind. Fly contact observations were made outside in a shady area, and two entomological field workers tallied fly-eye, -nose and -mouth contacts. The face was filmed (using a tripod-mounted Nikon D7200 single-lens reflex camera) for the 10-min observation period. We defined (1) fly-eye contact as a fly touching the eye, lid edge or eyelashes; (2) fly-nose contact as a fly touching the nostril area; (3) fly-mouth contact as a fly touching the lips or lip margin. Repeat contacts by the same fly were recorded as new contacts, as differentiating between new and repeat contacts was not possible.

**Fly capture.** Flies landing on individual's faces were caught, killed (by exposure to acetone vapour), and transported on ice to our local laboratory as previously described [7]. The observed child sat on a chair and for 15 minutes, or until 10 nets were used, flies landing on the child's face were disturbed and caught in a sterile net. Nets for catching flies (Bugdorm 38cm 300 μm mesh) were sterilised before use using Virkon-S (Aston Pharma, London; 1:100 in tap water), known to effectively remove DNA, and then hung in the sun for further UV sterilisation.

**Fly identification and testing for Ct DNA (whole flies).** In the laboratory, flies were identified with keys [16,17] as *M. sorbens*, *M. domestica*, 'other' (clearly another species) or 'unknown' (unidentifiable) using dissecting microscopes (S-20-2L x20 Stereo Microscope, Optika; MX4T stereomicroscope, Brunel). *Musca sorbens* and *M. domestica* were sexed. Flies were transferred individually into sterile tubes using forceps; a negative control was taken after processing each child's flies by wiping a sterile dacron swab (Puritan) around the petri dish used during identification. All dissecting equipment was sterilised before and between use using Virkon-S. Flies were temporarily stored at -20°C before transfer to a -80°C freezer, where they were stored until testing. Up to five flies per child were separately tested 'whole' by Ct qPCR; where more than five flies were captured (as in almost all instances), the first five that were stored for each child were chosen.

**Fly testing for Ct DNA (body parts).** Of 13 ocular positive households identified in the survey, entomological studies were done in 12 (due to entomological staff shortages during the testing round of the 13th house). In these 12 households only, flies remaining after testing five per child 'whole', were sectioned into three 'body parts' for testing: (1) head, (2) legs, (3) body (thorax, including wings, and abdomen). Therefore, up to five were processed in this way for two individuals per household. Flies were thawed for dissection, then fly parts were removed into separate sterile tubes in a microbiological safety cabinet, with negative controls performed as above. Fly parts were then stored at -20°C until testing. Dissecting equipment was sterilised with bleach before and between use. Body parts were separately tested by Ct qPCR.

**DNA extraction, quantification and load estimation.** Extraction and Ct DNA quantification in flies were performed as previously described [7], but in brief: flies were incubated in a digestion buffer (200μL; containing 1% Proteinase K) for 30 minutes at 37°C and two minutes at 95°C prior to extraction. Extractions were performed using the Biochain Blood and Serum kit (AMS Biotechnology Europe Ltd) and following manufacturer's recommendations. DNA was eluted using TE-buffer (80μL), stored at -80°C and thawed once for Ct detection. No flies or fly parts were bleached prior to testing. An in-house multiplex quantitative PCR (qPCR) assay was used for Ct detection, performed on a Quantstudio 7 flex Real-Time PCR machine (Applied Biosystems). This assay targets the Ct chromosomal *omcB* gene, plasmid pORF2 gene and human *RPP30* gene. Eluates were spiked with human DNA obtained from a Hep-2 cervical cell line prior to testing, to detect possible inhibition and to provide an internal

control (human *RPP30* gene). Samples were classed as Ct-positive after amplification of the *omcB* or pORF2 target within 40 cycles, and load estimation estimated via comparison with standards of known concentration [7].

## Laboratory fly feeding experiments: *Chlamydia trachomatis* DNA

**Fly preparation and feeding.** Internal and external recovery of Ct DNA was tested using six female flies (experiment 1), six male flies (experiment 2), and six males/six females (experiment 3). *Musca sorbens* (broad frons [18], S1 Table), colonised in 2018, were reared at the London School of Hygiene & Medicine according to previously published methods [19]. Flies of known and uniform age were starved of food (milk/sugar) and water prior to feeding experiments (experiment 1, 5h; experiment 2, 5h; experiment 3, 17.5h [overnight]). After starving, surviving flies were put into individual containers (60mL; Sterilin, UK) covered by a square of mesh (Nylon dress net, John Lewis, UK) with a small hole in the middle plugged with cotton wool.

Ct culture lysate for feeding experiments was generated by growing Ct A2497 in Hep-2 cervical epithelial cells and harvesting as described previously [20]. The number of chlamydial bodies was determined by measuring the number of copies of *omcB*, a single copy target, by qPCR.

In a microbiological safety cabinet, flies were fed aliquots of Ct culture (culture lysate containing EB at approximately 1000 EB/μL) by pipette through the hole in the mesh. For experiments 1 and 2, Ct suspension (10 μL) was added to each pot and flies given five minutes to consume the liquid before being transferred to a pot (Sterilin, UK). Three flies were 'knocked down' immediately (killed by insertion into an ice box), and three were knocked down after 60 minutes. For experiment 3, four aliquots of Ct suspension (2 μL) were added to each pot and the number of aliquots consumed counted. After feeding, flies were immediately transferred to sterile pots. Three female and three male flies were knocked down immediately, and three female and three male flies were knocked down after 60 minutes. Control flies were knocked down without feeding.

**Fly samples for *Ct* testing.** Knocked-down flies were first washed to remove the Ct feed from external surfaces. Individual flies were vortexed for 30s in phosphate-buffered saline (PBS)–0.5% Tween 20 (300μL) to generate the 'wash' sample. Flies were then transferred using sterile forceps into bleach (1ml, 1000 ppm) for 14m to remove any remaining surface DNA, after which the bleached carcass was re-washed by vortexing in PBS–0.5% Tween 20 (300μL). The fly head of the bleached carcass was then cut from the body using disposable scalpels in one smooth cut, ensuring no contact between the two samples, which were then put in separate tubes for testing. This generated the 'head' and 'body' samples, the latter including thorax, wings, legs and abdomen. All dissection equipment was sterile and care was taken to avoid cross-contamination between body parts. In experiment 3, fly carcasses were additionally dried (using a hairdryer) after the post-bleach wash, to further reduce cross-contamination between head and body samples. All samples were stored immediately in sterile tubes at -20˚C. DNA extraction, PCR quantification and load estimation was performed as for the field samples.

## Scanning electron microscopy

Scanning electron microscopy was used to investigate how Ct EB may adhere to the fly ultrastructure, specifically on the proboscis, feet or setae. Flies with minimal external (environmental) contamination were generated under aseptic conditions as follows. Puparia were placed into a 1% v/v bleach (CleannFresh Thick Bleach original, Robert McBride Ltd, UK) solution for one minute. Puparia were then transferred into milipore-filtered water for at least one minute, before being individually and aseptically transferred into sterile containers (60mL;

Sterilin, UK) for emergence. On the day of emergence, glucose solution was offered once (filter-sterilised through a 0.45 μm membrane filter, added to pot aseptically [10 μL]), thereafter flies were starved. The following day flies were fed aliquots (10 μL) of Ct culture (lysate, as above) at one of three concentrations (high, 25000 EB/μL; medium, 2500 EB/μL; low, 250 EB/μL) in a microbiological safety cabinet. Some flies were instead fed 2SP (Ct storage solution) as controls. Flies were observed for feeding, the approximate volume consumed was noted, then flies were knocked down on ice and transferred to individual vials of fixative (2.5% glutaraldehyde, 1% paraformaldehyde in Sorensen's Phosphate Buffer). Fixed specimens were dehydrated in a graded alcohol series before critical point drying with a Leica CPD300. Whole flies were mounted on stubs, coated with Gold-palladium and viewed with a Zeiss Ultraplus FEG scanning electron microscope.

## Statistical analysis

**Spatial distribution of Ct-positive flies.**   Household geolocation was electronically captured in encrypted forms using the open-source survey tool kit ODK Collect. Households were considered ocular positive if at least one resident was ocular positive, individuals were considered ocular positive if they had an ocular positive swab. For all households the distance in metres to the nearest ocular positive household was calculated (minimum distance), if a household itself was ocular positive, this distance was set as 0m. The association between household minimum distance and the percentage of Ct-positive flies caught at that household was estimated by a rate ratio obtained from a negative binomial regression, where the percentage of Ct-positive flies was the outcome and distance (in units of 100m) the primary exposure. Quadratic terms were tested in the model to assess for non-linear effects.

**Fly-eye contact.**   The association between fly-eye contact and person variables (disease: trachomatous inflammation-follicular [TF] or trachomatous inflammation-intense [TI]), Ct detected on flies caught from that person, ocular/nasal discharge, age, sex, bodyweight and ear temperature) and environmental variables (temperature, humidity, light intensity, wind and rain) was estimated by a rate ratio obtained from a negative binomial regression. The number of fly-eye contacts experienced in ten minutes was the outcome and variables were tested individually having adjusted for clustering at the household level. Results are presented as minimally adjusted (all adjusted for age only) and maximally adjusted (adjusted for all potential confounders associated with a substantial [~10%] change in effect estimate).

To test whether the association between disease (TF/TI) and fly-eye contact varied according to age, an interaction term between age group and disease was included in the negative binomial regression. Stratum specific rate ratios were presented.

Videos were made of each ten-minute fly-eye sample. Videos were initially analysed retrospectively for the summed duration of fly to eye visits, however, in a region of such high fly density this was quickly found to be impractical. Videos were instead retained as a back-up for fly-eye sampling.

**Ct carriage by flies/laboratory reared flies.**   We tested whether the recovered load (*omcB* copies/μL) varied by fly body part, fly sex, and time point (0/60 mins) using a linear regression with (log) *omcB* load as the outcome variable.

## Results

### Field studies of transmission dynamics

**Tested flies (caught from eyes).**   In total across the household survey, 3996 flies were caught from the eyes of 494 individuals aged 1 to 50 years living in 220 households (79% individuals aged 9 years or less). A mean of 9.7 flies (SD 1.21) were caught per individual. Most

flies caught were *M. sorbens* (90.1%; 3601), 5.4% (215) were *M. domestica*, 0.83% (33) were unidentified species and 3.7% (146) were unidentifiable or squashed. Overall, 81.2% (3243) of flies were female, 18.4% (736) male, and 0.4% (17) of unidentifiable/unknown sex. Of the *M. sorbens*, 82.2% (2961) were female, 17.5% (631) male and 0.25% (9) of unidentifiable/unknown sex; overall caught flies were 74.1% female *M. sorbens*. Of the *M. domestica*, 93% (200) were female and 6.9% (15) were male.

**Spatial distribution of Ct-positive flies.**   A total of 2053 flies were tested 'whole' (as entire flies) for Ct DNA. Overall, 2.6% (53) were Ct-positive, of these 81.1% (43) and 11.3% (6) were female and male *M. sorbens* respectively. One female *M. domestica* was Ct-positive, as were 3 unidentifiable flies. In ocular positive households, 22.4% (24/107) of caught flies were Ct-positive, while in ocular negative households 1.5% (29/1946) of caught flies were Ct-positive (Table 1). Overall, we found evidence that in ocular positive households the proportion of caught flies that were Ct-positive was higher than in non-ocular positive households (rate ratio [IRR] 15.12, 95% confidence interval [CI] 0·96–238.26, $P = 0·05$), however, the confidence interval was very wide due to the small number of ocular positive households. Of the 2053 tested flies, 1964 were from 208 households with geospatial data (latitude/longitude). Through mapping, we saw that households with Ct-positive flies tended to cluster nearby to ocular positive households (Fig 1), with a rapid drop-off in fly positivity over the first 200m. For each 100m increase in distance from an ocular positive household, we observed a reduction in the proportion of Ct-positive flies of about a quarter (RR 0.74, 95% CI 0.63–0.87, $P<0.001$; Fig 2).

**Fly-eye contact.**   Overall, individuals experienced approximately twice as many fly-eye contacts as -nose or -mouth contacts (mean 43.9, 24.0 and 25.6; standard deviation [SD] 39.2, 27.5 and 24.1 respectively). Contacts decreased as the study progressed (from hot to rainy season), with mean fly-eye contacts of 53.9, 49.7 and 32.1 (SD 42.7, 41.0, 31.7) in April, May and June respectively.

There was evidence that fly-eye contact was higher on individuals with TF/TI (rate ratio [RR] 1.42, 95% confidence interval [CI] 1.18–1.70, $P<0.001$, Fig 3A), and this effect remained after adjusting for both age and ocular discharge (RR 1.38, 95% CI 1.15–1.66, $P = 0.001$) (Table 2). The evidence of an association between ocular discharge and fly-eye contact was weaker, after adjusting for disease and age there was little evidence to suggest that ocular discharge alone was associated with fly-eye contact (RR 1.15, 95% CI 0.97–1.35, $P = 0.11$).

Among uninfected children, as age increased, fly-eye contact decreased (Table 3 and Fig 3B). However, among infected children, as age increased fly-eye contact remained at the higher levels experienced by the youngest children. In the youngest children, aged 1–3 years,

**Table 1. Fly positivity in (Ct) ocular positive (at least one Ct-positive ocular swab among household members) and ocular negative households.**

|  | Ocular pos | Ocular neg[A] | Total |
|---|---|---|---|
| Total households | 12 | 204 | 216 |
| Total individuals sampled for flies | 23 | 402 | 425 |
| Age of sampled individuals (median [IQR], range) | 7 (5–10)[B], 1–20 | 7 (4–9), 1–50 | 7 (4–9) |
| Total flies tested | 107 | 1946 | 2053 |
| Positive flies (% total) | 24 (22.4%) | 29 (1.5%) | 53 (2.6%) |
| omcB load (copies/mL; median [IQR]; $n$[C]) | 14.3 (2.5–55.1), 23 | 1.7 (0.5–15.6), 20 | 5.2 (0.8–54.7), 43 |

[A]Only 196 households with geospatial information

[B]$n = 21$, age data missing for two individuals

[C]$n$ = sample size for omcB load (excluding omcB negative but plasmid positive samples)

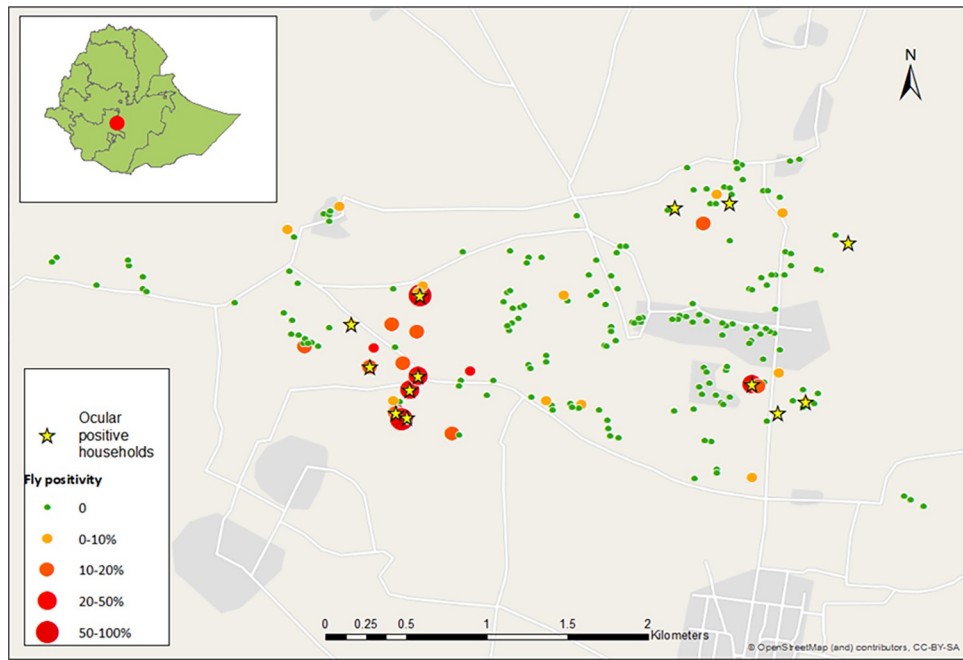

**Fig 1. Ocular positive households and *M. sorbens* positivity in ocular negative households across the study site.** Of 247 households in the population-based survey, flies were tested in 196 ocular negative households with a geolocation, and in 12 of 13 ocular positive households. A mean of 9.5 flies were tested per ocular negative household (SD 1.3), and a mean of 8.9 (SD 2.5) flies were tested per ocular positive household. (Base map data from OpenStreetMap).

there was little difference in fly-eye contact between infected and uninfected children. In 4-6- and 7–9-year-olds, those with infection experienced more fly-eye contacts than those without (Table 3 and Fig 3B). However, the evidence for effect modification was very weak ($P = 0.12$).

Increasing age (unadjusted) was associated with decreased fly-eye contact ($P<0.001$), but not when also adjusting for disease, ocular discharge, and weight ($P = 0.77$; Table 2). However, per kg increase in bodyweight was strongly associated with a very small decrease in fly-eye

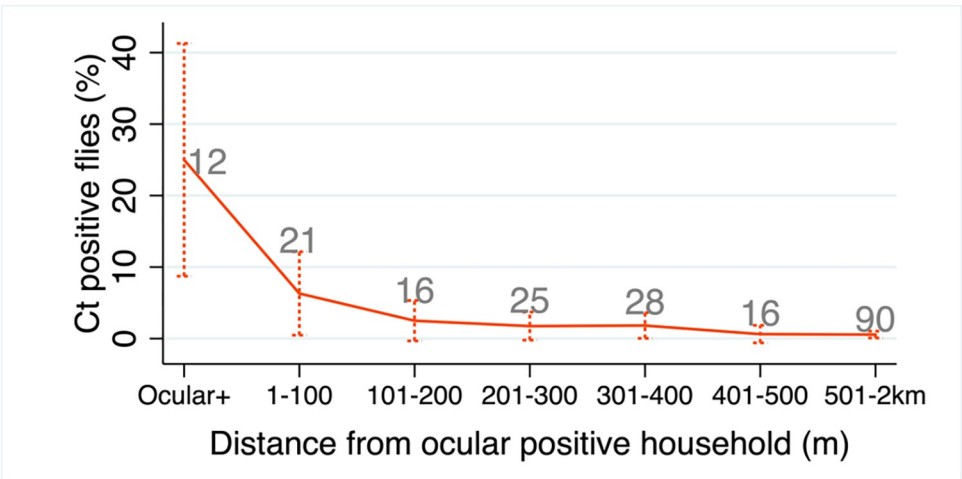

**Fig 2. *Musca sorbens* positivity for Ct DNA over distance from the nearest ocular positive household.** Mean +/-95% confidence intervals shown; sample size (households) given above data points.

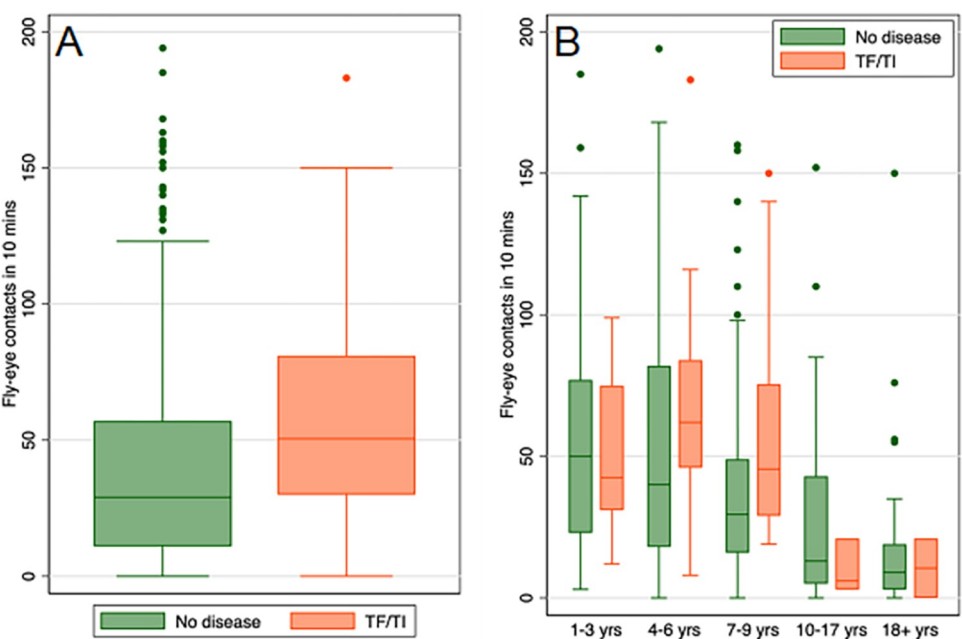

**Fig 3. Fly-eye contact experienced by individuals with and without TF/TI (A) and additionally in different age groups (B)**. Boxes show median (interquartile range) and outliers (points). Sample sizes are (A) no disease, 346; disease, 70 (B) 1–3 years, no disease, 45; disease, 20; 4–6 years, no disease, 103; disease, 25; 7–9 years, no disease, 118; disease, 20; 10–17 years, no disease, 41; disease, 3; 18+ years, no disease, 39; disease, 2.

contact (un-adjusted and adjusted for age, ocular discharge and disease; Table 2). Per degree Celsius increased body (ear) temperature was associated with a 15–20% increase in fly-eye contact (un-adjusted and adjusted; Table 2). The presence of wind was associated with fewer fly-eye contacts, and there was some evidence that increased light intensity was associated with very small decreases in fly-eye contact (Table 2).

## Ct carriage by flies

**Field-caught flies.** A total of 106 flies from 23 individuals in the ocular positive households were cut into body parts before testing for Ct. Of these flies, 22.6% (24/106) were Ct-positive, and the majority (19/24) were female *M. sorbens* (Table 4, upper panel). All 24 Ct-positive flies had Ct-positive bodies, and of these, five also had Ct-positive heads and legs (Table 4, lower panel). One Ct-positive fly was unidentifiable or unknown species, all others were *M. sorbens*. The median *omcB* load in bodies was higher than that recovered from heads or legs, although all were very low (bodies, 2.4 copies/µL (interquartile range, IQR: 0.5–12.3); heads 0.3 copies/µL (IQR: 0.3–1.3); legs 0.9 copies/µL (IQR: 0.5–1.0)).

**Laboratory reared flies.** Ct DNA was recovered from all three sample sites (fly wash, head and body) at both time points (0 and 60 minutes) in all three experiments (Fig 4). The fly wash would contain any Ct recovered from any part of the exterior surfaces of the fly. Because the fly carcass was then bleached for 14 minutes, any remaining DNA on the exterior of the fly was destroyed. Testing the remaining two samples ('head' and 'body') would therefore reveal Ct DNA only inside those body parts, as all exterior DNA was destroyed in the bleach wash. Taking timepoints T0 and T60 together, we recovered 3.86 times more *omcB* copies from the fly body relative to the fly head (95% CI 0.23–11.82, *P* = 0·02) and 3.10 times more *omcB* copies from the fly wash relative to the fly head (95% CI 1.01–9.49, *P* = 0·05). There was no difference

**Table 2. Association between fly-eye contacts and measured variables in 438 individuals.** Raw data given as well as rate ratios of fly contact relative to baseline. SD = Standard Deviation, n = number of participants, IQR = Interquartile Range.

| | | | n | Mean contacts (SD) | Median contacts (IQR) | Minimally adjusted[A] RR (95% CI) | P-value | Maximally adjusted[B] RR (95% CI) | P-value |
|---|---|---|---|---|---|---|---|---|---|
| **Fly-eye contact** | | | | | | | | | |
| Infection or disease variables | Presence of TF/TI | no | 346 | 41.03 (39.26) | 29 (11–57) | baseline | | baseline | |
| | | yes | 70 | 57.19 (39.20) | 50.5 (30–81) | 1.42 (1.18–1.70) | <0.001 | 1.38 (1.15–1.66)[C] | 0.001 |
| | Either eye Ct positive | no | 403 | 42.73 (38.46) | 32 (13–62) | N/A | | N/A | |
| | | yes | 13 | 75.31 (49.51) | 72 (46–83) | | | | |
| | Total number of individuals in household with an ocular positive swab | 0 | 405 | 42.68 (38.33) | 32 (13–62) | N/A | | N/A | |
| | | 1 | 14 | 56.79 (54.89) | 38.5 (20–77) | | | | |
| | | 2 | 3 | 64.0 (18.52) | 63 (46–83) | | | | |
| | | 3 | 6 | 83.83 (40.34) | 73.5 (60–110) | | | | |
| | At least one fly caught from this person Ct positive | no | 362 | 43.96 (39.37) | 32 (15–63) | baseline | | baseline | |
| | | yes | 32 | 50.69 (44.15) | 39 (21.50–69) | 0.99 (0.74–1.31) | 0.93 | 0.98 (0.74–1.30)[C] | 0.9 |
| | Ocular discharge | no | 278 | 38.39 (35.26) | 28 (9–57) | baseline | | baseline | |
| | | yes | 138 | 54.56 (44.32) | 37.5 (21–82) | 1.20 (1.02–1.41) | 0.03 | 1.15 (0.97–1.35)[D] | 0.11 |
| | Nasal discharge | no | 259 | 39.20 (39.27) | 25 (9–57) | baseline | | baseline | |
| | | yes | 157 | 51.26 (38.02) | 40 (25–67) | 1.13 (0.95–1.35) | 0.16 | 1.09 (0.92–1.30)[D] | 0.32 |
| Person variables | Age (years) | 1–3 | 65 | 56.77 (39.33) | 49 (28–77) | baseline[H] | | baseline[I] | |
| | | 4–6 | 128 | 54.86 (43.13) | 45 (21.5–83) | 0.86 (0.71–1.06) | 0.15 | 1.06 (0.87–1.29)[C,D,E] | 0.57 |
| | | 7–9 | 138 | 40.49 (33.36) | 32.50 (19–53) | 0.68 (0.55–0.83) | <0.001 | 0.99 (0.80–1.24) | 0.96 |
| | | 10–17 | 44 | 27.84 (34.40) | 12.50 (4.5–42) | 0.37 (0.27–0.50) | <0.001 | 0.88 (0.60–1.28) | 0.5 |
| | | 18+ | 41 | 16.51 (26.95) | 9 (3–19) | 0.28 (0.20–0.39) | <0.001 | 1.04 (0.64–1.69) | 0.87 |
| | Sex | Female | 212 | 42.65 (39.16) | 32 (11.5–63) | baseline | | | |
| | | Male | 204 | 44.90 (39.29) | 32.5 (18.5–62.5) | 0.97 (0.84–1.11) | 0.64 | 0.95 (0.83–1.09)[C,D,E] | 0.5 |
| | Bodyweight, continuous | | | | | 0.96 (0.95–0.97) | <0.001 | 0.96 (0.95–0.97)[C,D] | <0.001 |
| | Tympanic temp, continuous | | | | | 1.20 (1.11–1.31) | <0.001 | 1.16 (1.07–1.26)[C,D,E,F] | <0.001 |
| Environmental variables | Ambient temp (oC), continuous | | | | | 1.0 (0.98–1.02) | 0.77 | 1.01 (0.97–1.04)[G] | 0.8 |
| | Rel. humidity (%), continuous | | | | | 1.0 (0.99–1.00) | 0.57 | 1.0 (0.99–1.01)[G] | 0.85 |
| | Light intensity (lux/100), continuous | | | | | 0.99 (0.99–1.0) | 0.01 | 0.99 (0.99–1.0)[G] | 0.01 |
| | Presence of wind | no | 146 | 53.24 (41.54) | 46.50 (17–77) | baseline | | baseline | |
| | | yes | 282 | 39.01 (37.04) | 28 (11–55) | 0.74 (0.64–0.87) | <0.001 | 0.76 (0.65–0.89)[G] | 0.001 |
| | Presence of rain | no | 410 | 43.75 (38.71) | 32.50 (15–63) | N/A | | N/A | |
| | | yes | 18 | 46.39 (49.78) | 28.50 (4–80) | | | | |

[A]Adjusted for age only (excluding age)

[B]All adjusted for age (excluding age), with additional adjustments annotated

[C]Adjusted for ocular discharge

[D]Adjusted for disease (TF/TI)

[E]Adjusted for weight

[F]Adjusted for ambient temperature

[G]Adjusted for all other environmental variables (excluding rain)

[H]P-value testing overall association between fly-eye contact and age <0.001

[I]P-value testing overall association between fly-eye contact and age 0.77

N/A Statistical analysis not appropriate

**Table 3. Stratum-specific predictions for fly-eye contact according to categories of TF/TI disease and age, with and without adjustment for ocular discharge.** Categories in panel A are age groups, categories in panel B are disease (No–No TF/TI; Yes–Presence of TF/TI).

| | Strata | Category | n | No adjustment | | Adjusted for ocular discharge | |
|---|---|---|---|---|---|---|---|
| | | | | RR (95% CI) | P-value[A] | RR (95% CI) | P-value[B] |
| | Presence of TF/TI | Age 1–3 years | 20 | baseline | | baseline | |
| | | Age 4–6 years | 25 | 1.23 (0.85–1.76) | 0.27 | 1.24 (0.86–1.77) | 0.24 |
| | | Age 7–9 years | 20 | 0.97 (0.65–1.45) | 0.89 | 1.0 (0.67–1.50) | 1.00 |
| | | Age 10–17 years | 3 | 0.24 (0.08–0.70) | 0.01 | 0.26 (0.09–0.74) | 0.01 |
| A | | Age 18+ years | 2 | 0.14 (0.02–0.91) | 0.04 | 0.15 (0.02–0.97) | 0.05 |
| | No TF/TI | Age 1–3 years | 45 | baseline | | baseline | |
| | | Age 4–6 years | 103 | 0.81 (0.63–1.02) | 0.08 | 0.83 (0.65–1.06) | 0.14 |
| | | Age 7–9 years | 118 | 0.65 (0.51–0.83) | <0.001 | 0.68 (0.53–0.87) | <0.01 |
| | | Age 10–17 years | 41 | 0.38 (0.27–0.53) | <0.001 | 0.40 (0.28–0.56) | <0.001 |
| | | Age 18+ years | 39 | 0.29 (0.20–0.41) | <0.001 | 0.30 (0.21–0.43) | <0.001 |
| | Age 1–3 years | No | 45 | baseline | | baseline | |
| | | Yes | 20 | 1.12 (0.79–1.58) | 0.52 | 1.11 (0.78–1.57) | 0.56 |
| | Age 4–6 years | No | 103 | baseline | | baseline | |
| | | Yes | 25 | 1.71 (1.3–2.25) | <0.001 | 1.65 (1.25–2.18) | <0.001 |
| B | Age 7–9 years | No | 118 | baseline | | baseline | |
| | | Yes | 20 | 1.67 (1.22–2.29) | 0.001 | 1.62 (1.18–2.23) | <0.001 |
| | Age 10–17 years | No | 41 | baseline | | baseline | |
| | | Yes | 3 | 0.70 (0.25–2.01) | 0.51 | 0.71 (0.25–2.02) | 0.52 |
| | Age 18+ years | No | 39 | baseline | | baseline | |
| | | Yes | 2 | 0.55 (0.08–3.58) | 0.53 | 0.55 (0.08–3.59) | 0.53 |

[A]P value testing if the overall effect of disease differs by age = 0.09

[B]P value testing if the overall effect of disease differs by age = 0.12

**Table 4. Upper panel: 106 flies from (Ct) ocular positive households were 'dissected' into body parts; head, legs and 'body' (thorax/abdomen/wings).** Percentages, and fraction, given in all cells. Lower panel: flies from ocular positive households were cut into body parts; head, legs and 'body' (thorax/abdomen/wings) and tested separately for Ct. Salmon shaded cells indicate positivity for Ct.

| Sample | *M. sorbens* (% positive; +/total) | *M. domestica* (% positive; +/total) | unknown spp. (% positive; +/total) | Total (% positive; +/total) |
|---|---|---|---|---|
| Flies | 23.7 (23/97) | 0% (0/4) | 20% (1/5) | 22.6% (24/106) |
| Females | 29.2 (19/65)[A] | 0% (0/3) | 50% (1/2) | 28.6% (20/70)[A] |
| Males | 13.3 (4/30)[A] | 0% (0/1) | 0% (0/3) | 11.8% (4/34)[A] |
| Heads | 6.3 (6/96)[B] | 0% (0/4) | 0% (0/5) | 5.7% (6/105)[A] |
| Legs | 6.2 (6/97) | 0% (0/4) | 0% (0/5) | 5.7% (6/106) |
| Bodies | 23.7 (23/97) | 0% (0/4) | 20% (1/5) | 22.6% (24/106) |

| Fly | 1 | 2 | 3 | 4 | 5 | 6 | 7 | 8 | 9 | 10 | 11 | 12 | 13 | 14 | 15 | 16 | 17 | 18 | 19 | 20 | 21 | 22 | 23 | 24 |
|---|---|---|---|---|---|---|---|---|---|---|---|---|---|---|---|---|---|---|---|---|---|---|---|---|
| Body | ■ | ■ | ■ | ■ | ■ | ■ | ■ | ■ | ■ | ■ | ■ | ■ | ■ | ■ | ■ | ■ | ■ | ■ | ■ | ■ | ■ | ■ | ■ | ■ |
| Head | | | | ■ | | ■ | ■ | | | | | | ■ | | | | | | | | ■ | | | |
| Legs | | | | | | ■ | ■ | ■ | | | ■ | | ■ | | | | | | | | ■ | | | |

[A] Two unknown sex *M. sorbens*

[B]One head lost during processing

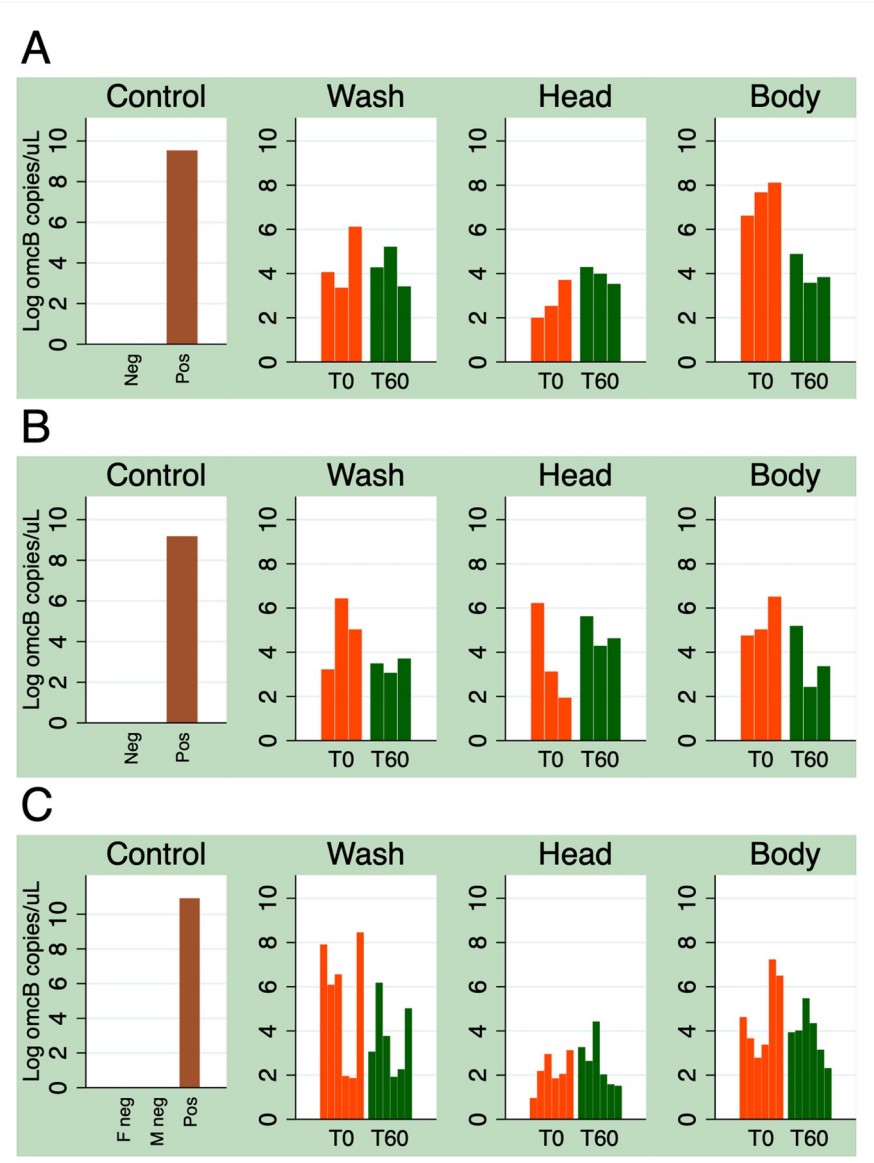

**Fig 4. Ct DNA recovered from individual laboratory-reared *M. sorbens* fed Ct in solution.** Flies were either knocked down immediately after feed (T0) or in one hour (T60). Flies in (A) were female and (B) were male, all were given access to 10 μL Ct suspension (1000 omcB copies/μL) for five minutes. Flies in (C) were 6 males/6 females; bars ordered as 3 females then 3 males at each timepoint.

in the number of *omcB* copies recovered from male or female flies ($P = 0.4$). Across all samples, 0.39 times fewer *omcB* copies were recovered after 60 minutes than at T0 (95% CI 0.15–0.97, $P = 0.04$).

**External carriage of Ct by *M. sorbens*.**   Of eight flies fed Ct culture lysate, EB were observed on only one fly, fed at the lowest concentration (approximately 250 EB/μL) (Fig 5). EB were observed on the setae of one leg, adhering in clumps to individual or groups of hairs. EB were seemingly absent from the sticky tenent setae of the pulvillus, or any site on the proboscis or the pseudotrachae of the proboscis.

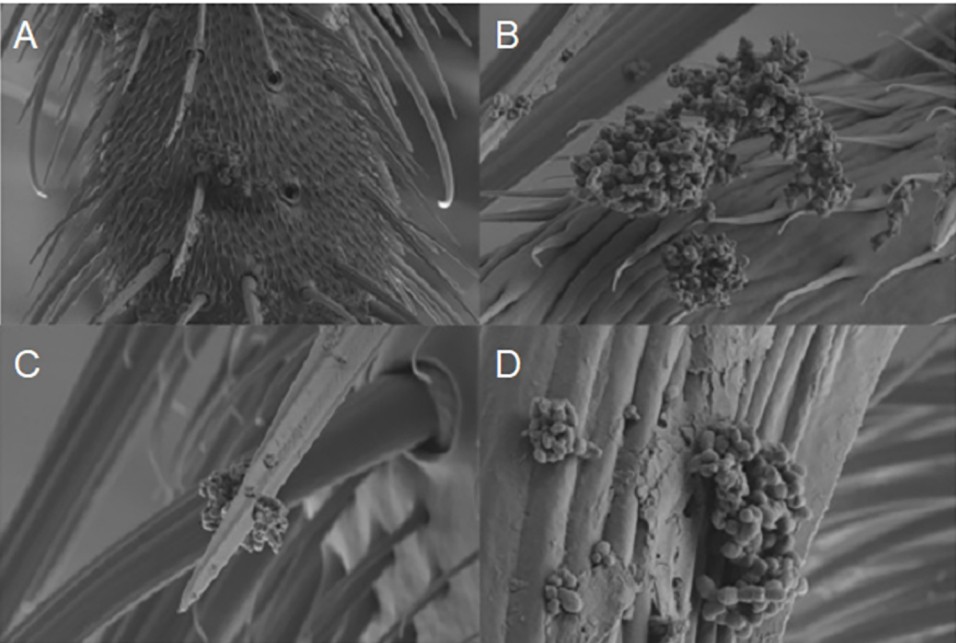

**Fig 5.** (A)-(D). **Ct culture lysate, containing EB, can be seen clinging to the hairs on *M. sorbens* legs.**

## Discussion

In the 12 ocular positive households, almost one quarter of caught flies were Ct-positive (flies on which Ct DNA was detected), relative to only 1.5% in ocular negative households. By mapping the distribution of flies carrying Ct across the study site, we observed that Ct-positive flies cluster around ocular positive households. For each 100m increase in distance from an ocular positive household, we observed a reduction in the proportion of Ct-positive flies of about a quarter. Of person-level variables, there was evidence that having trachoma (TF/TI disease), lighter bodyweight, or increased body temperature was associated with increased fly-eye contact. Of environmental variables the presence of wind was associated with fewer contacts and increasing light with a very small decrease in contacts. All 24 Ct-positive field-caught flies that were separated into body parts for testing had Ct-positive 'bodies' (thorax and abdomen). Five were additionally Ct-positive for both legs and heads, one additionally positive for legs and another for head. All laboratory reared, Ct-fed flies were found to be Ct-positive for Ct DNA at all three sites tested: the exterior surface (fly 'wash'), head (bleached) and body (thorax including wings/legs and abdomen, bleached). Less Ct DNA was recovered from heads than from bodies or exterior (wash), and there was some evidence that the amount of Ct DNA recovered decreased after 60 minutes across all sites. Scanning electron microscopy of Ct-fed *M. sorbens* demonstrated how Ct bodies can adhere to the setae and hairs on the body.

Few studies have tested wild-caught *M. sorbens* for Ct DNA [6,21,22]. Of those, total numbers of tested flies have been significantly lower than those reported in the current study. Furthermore, no other study has examined rates of positivity according to proximity to households with ocular positive individuals. Our finding of Ct-positive flies clustering within and close to ocular positive households is significant for several reasons. Trachoma is known to cluster geospatially (at district [23,24], village [7,23,25,26], household [27–30] and bedroom [31] level). Our spatial data may indicate that either flies in this locality do not travel far, or that Ct contamination of flies is cleared before the flies have migrated far. In the absence of

pathogen replication/amplification within the vector, as is assumed for Ct in *M. sorbens*, a decrease in bacteria loads on the exterior of the fly is inevitably caused by exposure (UV, heat) and/or interspecific competition from the microorganisms colonising the flies' exterior (part of the flies' microbiome). Similarly, bacterial loads would decrease internally due to digestion, the flies' immune response and interspecific competition from the microbiome of the gut [8]. That flies are not found far from ocular positive households is consistent with the maintenance of (vector-borne) focal trachoma and the possibility that flies may perpetuate transmission within clusters.

After controlling for age, we observe an association between increased bodyweight and decreased fly-eye contacts. We do not observe an association between age and contacts having controlled for weight, disease and discharge. This is indicative that weight is important to fly contacts, with the lightest children experiencing the most contacts. In a previously reported study we observed that contacts decreased as age increased [32]. Those with lower bodyweight (children) are more likely to have ocular and nasal discharge, although here, the effect of bodyweight remains after controlling for discharge. It is thought that female flies are attracted to this protein-rich food source to support egg production; [6] as previously documented we observe a significant skew towards female eye-seeking flies [6,33]. Active trachoma, Ct infection, and Ct loads are higher in children, [1], and they serve as the principal reservoir of Ct in the community [6]. Here we measure several interrelated variables: disease, discharge, bodyweight/age and fly-eye contact. Further, fly-eye contact from disease vectors increases infection exposure. Discerning causal pathways from observational studies may not be possible.

An association between disease and increased vector nuisance/aggression may be anticipated. It is known from other vector-borne disease systems that infection alone (in asymptomatic individuals) can lead to increased vector contact via host manipulation mechanisms [34–36], for example modification of the host odour profile [37–39]. Overall, the youngest children experienced the most fly-eye contact. However, while in uninfected children fly-eye contact rates dropped steadily with age, in infected children fly-eye contact rates remained at the higher rates until approximately nine years of age. These data give some indication that in the age range of four to nine years, Ct infection may contribute to increased fly-eye contact rates. However, the study had limited power to detect effect modification. Given the estimated difference in the magnitude of the effect, it could warrant further investigation in future studies.

All field-caught flies that were tested as separate body parts (head, legs and body) had a Ct-positive body with only a few having a Ct-positive head or legs. Low median loads were recovered from all samples in the field flies, with Ct amounts approaching the test limit of detection [40]. Body samples were the largest in size, which may have allowed Ct to be more often recovered. Furthermore, field samples underwent a freeze-thaw cycle that may have caused smaller and low Ct-load samples to become undetectable. This could mean that more heads and legs were positive than we were able to detect. Fly bodies were not surface bleached, so the Ct DNA cannot be located to the exterior or interior of the fly. Previous studies have indicated that the most significant carriage of pathogens by mechanical vectors, in terms of numbers and duration, will be within the alimentary canal [8]. Ct may be ingested in greater amounts than that collected externally via contact, and pathogenic bacteria have typically been found to survive longer internally than on the surface of a fly [8]. This has been attributed to factors including fly cleaning, desiccation due to fly surface temperatures, and insolation and UV radiation from the sun. We can speculate that the Ct isolated from the flies' bodies was, therefore, present in the alimentary canal, but further studies are required to investigate this. Future studies of field-caught flies should include aseptic dissection and testing of gut regions, alongside exterior washes for surface Ct, and appropriate bleaching to avoid exterior-interior DNA contamination.

In our laboratory feeding experiments we recovered the greatest loads of Ct from the exterior wash and the bleached body samples (thorax, legs, wings and abdomen) relative to the bleached head. The body sample represents Ct recovered from the alimentary canal as this is the only place that Ct could be localised internally (after ingestion). Therefore, our laboratory studies indicate that following ingestion Ct can be recovered in higher amounts from the alimentary canal and exterior surfaces of the entire fly relative to the interior of the head (within the esophagus). Therefore, as with the field flies, more Ct was recovered from the largest samples. These laboratory feeding experiments demonstrate that after feeding, Ct DNA is found at all sites for at least 60 minutes, as may be expected following exposure to an appealing (sugary) meal of high-dose Ct culture. It would be interesting to conduct experiments testing how Ct loads persist at these sites over time, especially following exposure to environmental stressors like UV radiation as mentioned above. Future laboratory experiments would benefit from testing the recovery of Ct from flies at more biologically realistic loads as loads of Ct in ocular discharge are often low [7,41]. The improved relevance of such studies must be balanced against the challenges of measuring such small amounts; further laboratory studies would also benefit from greater replication that was possible within the constraints of this study.

The scanning electron images of Ct-fed flies demonstrate both how Ct EB can become caught in the dense hairs covering *M. sorbens*, and how exposed they are at that site. It is interesting that the only fly seen to carry Ct by SEM was one fed the lowest dose of Ct, 250 EB/μL. As all flies were observed to feed, any culture debris on the other flies may either have been cleaned off by the fly or EB may have been present individually and therefore harder to find during SEM.

All field studies described were conducted in a region of less than 5km$^2$ in Oromia. Transmission routes of Ct are multiple and thought to vary regionally according to the environment [14]. While our data are suggestive that flies may play a role in supporting transmission within and nearby to ocular positive households, studies were conducted in a dry season of relatively high fly density. Fly-borne transmission may be less important in seasons of lower fly density, or indeed regions with lower fly densities. Furthermore, while the data contribute significantly to our understanding of fly-borne trachoma and indicate several avenues for further research, sample sizes are small. In the field studies, the total number of Ct-carrying flies caught was low, and laboratory studies were conducted using small numbers of flies. More studies of this nature are needed.

Should further evidence corroborate these findings, our observation of Ct-carrying flies clustering close to ocular positive households may have significant application to disease control: control of fly populations or protection from fly-eye contact in high-prevalence clusters alone may be enough to mitigate against onwards fly-borne transmission, in this region at least. Calyptratae are noted to fly furthest when there is little to detain them [8]. In a different context therefore, perhaps where breeding site or host availability is scant, Ct-carrying *M. sorbens* may travel further. More options for the control of this important species should be considered, for example further insecticide-based methods, removal trapping, or personal protection. The 'body parts' experiments allude to field-caught flies harbouring greater amounts of Ct internally. These are preliminary, exploratory experiments and must be followed by aseptic dissection and testing of regions of the alimentary canal: i.e. the proventriculus, crop, midgut, hindgut and rectum, over a time series, to determine the temporospatial location of Ct post-ingestion. While interesting, published studies of this nature using *M. domestica* ultimately do not deepen our understanding of endemic *M. sorbens* trachoma transmission. Tests to indicate viability should be conducted, i.e. mRNA, viability PCR, or immunofluorescence methods [42]. It is also critical to explore onwards transmission routes from the fly: frequent regurgitation and defecation by filth flies allows the deposition of ingested

microbes including pathogens [43]. Vectorial competence, as described above, encompasses the ability of a vector to acquire and subsequently transmit the pathogen [13]. Broadly, this includes (1) uptake of the pathogen, (2) any replicative stage or process, (3) release of the pathogen into or onto a subsequent host. In the present paper we have only addressed the first of these processes, and all aspects of vectorial competence should be explored to thoroughly establish the role of *M. sorbents* in Ct transmission.

## Supporting information

**S1 Table. Head and frons measurements of ten males from our M. sorbens colony at the London School of Hygiene & Tropical Medicine.**
(DOCX)

## Acknowledgments

We are indebted to the participants of the field studies and their communities. We thank the wider team of Stronger-SAFE fieldworkers including Damitu Legese, Meseret Guye, Ewunetu Melese, Gadisa Deressa, Korso Hirpo, Rufia Kedir, Asanti Ahmed, Teka Ashagrie, Fitsum Shapa and Kibreab Beshano.

## Author Contributions

**Conceptualization:** Ailie Robinson, Bart Versteeg, Anna Last, Matthew J. Burton, James G. Logan.

**Data curation:** Ailie Robinson.

**Formal analysis:** Ailie Robinson, David Macleod.

**Funding acquisition:** Virginia Sarah, Anna Last, Matthew J. Burton.

**Investigation:** Ailie Robinson, Oumer Shafi Abdurahman, Innes Clatworthy, Gemeda Shuka, Dereje Debela, Gebreyes Hordofa, Laura Reis de Oliveira Gomes, Muluadam Abraham Aga, Gebeyehu Dumessa.

**Methodology:** Ailie Robinson, Bart Versteeg, Innes Clatworthy, David Macleod.

**Project administration:** Oumer Shafi Abdurahman.

**Supervision:** Anna Last, Matthew J. Burton, James G. Logan.

**Validation:** David Macleod.

**Writing – original draft:** Ailie Robinson.

**Writing – review & editing:** Ailie Robinson, Bart Versteeg, Oumer Shafi Abdurahman, Innes Clatworthy, Gemeda Shuka, Dereje Debela, Gebreyes Hordofa, Laura Reis de Oliveira Gomes, Muluadam Abraham Aga, Gebeyehu Dumessa, Virginia Sarah, David Macleod, Anna Last, Matthew J. Burton, James G. Logan.

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
