## [Decision Letter · Decision Letter 0]

17 Feb 2024

Dear Mr Macleod,

Thank you very much for submitting your manuscript "Further studies on the transmission of Chlamydia trachomatis by Musca sorbens in Oromia, Ethiopia" for consideration at PLOS Neglected Tropical Diseases. As with all papers reviewed by the journal, your manuscript was reviewed by members of the editorial board and by several independent reviewers. In light of the reviews (below this email), we would like to invite the resubmission of a significantly-revised version that takes into account the reviewers' comments, which are straight forward.

We cannot make any decision about publication until we have seen the revised manuscript and your response to the reviewers' comments. Your revised manuscript is also likely to be sent to reviewers for further evaluation.

Sincerely,

Aysegul Taylan Ozkan, M.D., Ph.D.,

Academic Editor

Amy Morrison

Section Editor

Reviewer's Responses to Questions

**Key Review Criteria Required for Acceptance?**

**Methods**

-Are the objectives of the study clearly articulated with a clear testable hypothesis stated?

-Is the study design appropriate to address the stated objectives?

-Is the population clearly described and appropriate for the hypothesis being tested?

-Is the sample size sufficient to ensure adequate power to address the hypothesis being tested?

-Were correct statistical analysis used to support conclusions?

-Are there concerns about ethical or regulatory requirements being met?

Reviewer #1: Yes, yes, yes, yes, yes, no concerns

Reviewer #2: (No Response)

Reviewer #3: (No Response)

**Results**

-Does the analysis presented match the analysis plan?

-Are the results clearly and completely presented?

-Are the figures (Tables, Images) of sufficient quality for clarity?

Reviewer #1: Yes, yes, yes

Reviewer #2: (No Response)

Reviewer #3: (No Response)

**Conclusions**

-Are the conclusions supported by the data presented?

-Are the limitations of analysis clearly described?

-Do the authors discuss how these data can be helpful to advance our understanding of the topic under study?

-Is public health relevance addressed?

Reviewer #1: Yes, yes, yes, yes

Reviewer #2: (No Response)

Reviewer #3: As provided on my comments

**Editorial and Data Presentation Modifications?**

Reviewer #1: Lines 39-40 : "that M. sorbens flies contribute to transmission of the disease”. Strictly speaking, the contribute to transmission of the infection.

42: please change “regions” to “populations” – “regions” has a political element, and the prevalence is found in a human population

47: “is the leading cause of infectious blindness worldwide” – please change to “is the leading infectious cause of blindness worldwide”: the blindness itself is not infectious

55. “the species ratio is always predominantly”, suggest change to “the species ratio of flies found at the eye is always predominantly”

139-140 and 148-149: the authors note that Virkon-S sterilized the nets and dissecting equipment, but does that solution degrade the DNA sequences of interest?

150: Five flies per child were… Should this read, “Up to five flies per child were…”?

160: this seems to contradict 148-149

196: Does the parenthetical text indicate that ice was used to kill the flies?

247 and 257: The words “The effect of…” implies causality, whereas I think all that’s being looked for here is an association

345: suggest change “ear (body)” to “body (ear)”

348: the word “in” is missing I think

419-420: “This is indicative that weight is important to fly contacts, with the lightest children experiencing the most contacts”. This suggests volition on the part of the flies. Was an analysis of this association done controlling for presence of active trachoma, which is also more likely to be found in smaller / lighter kids?

435: “The youngest children experienced the most fly-eye contact, but among infected children contact levels remained at this higher level for longer (until approximately nine years) than in uninfected children.” This sounds like the description of a nine-year-long longitudinal study, which I don’t think is what is intended.

Reviewer #2: (No Response)

Reviewer #3: (No Response)

**Summary and General Comments**

Reviewer #1: Nice study. I have just a few minor comments, indicated in the "editorial and data presentation" box above

Reviewer #2: Dear Editorial staff / Authors,

I have reviewed the manuscript PNTD-D-24-00098, entitled “Further studies on the transmission of Chlamydia trachomatis by Musca sorbens in Oromia, Ethiopia”; The manuscript seems well organized, but kindly you can find some points needs to be revised by the respectable authors, as below:

1- The type of study should be mentioned in the “methods” part.

2- The manuscript needs to be improved for English language.

3- In abstract it has been written “Fly-eye contact was positively associated with the presence of trachoma disease, lower body weight and increased body temperature.” But the body weight loss and fever have not been defined in this survey.

4- As mentioned in line 10: “the role of Chlamydia trachomatis as vectors has been convincingly demonstrated via randomised controlled trials in the Gambia”, why this study has been performed on a well-known fact?

5- In line 37 “The authors found evidence of on Ct on flies and…” the proposition “on” has been repeated twice.

6- Considering the Oromia region of Ethiopia as the place of research performing, and contribution of some respected authors outside Ethiopia, it seems better that the role of each author be specified in the manuscript.

Best regards,

Reviewer

Reviewer #3: (No Response)

PLOS authors have the option to publish the peer review history of their article (what does this mean?). If published, this will include your full peer review and any attached files.

Reviewer #1: No

Reviewer #2: No

Reviewer #3: No
---

## [Decision Letter · Decision Letter 1]

3 May 2024

Dear Mr Macleod,

Thank you very much for submitting your manuscript "Field- and laboratory-based studies on correlates of Chlamydia trachomatis transmission by Musca sorbens: determinants of fly-eye contact and investigations into fly carriage of elementary bodies" for consideration at PLOS Neglected Tropical Diseases. As with all papers reviewed by the journal, your manuscript was reviewed by members of the editorial board and by several independent reviewers. The reviewers appreciated the attention to an important topic. Based on the reviews, we are likely to accept this manuscript for publication, providing that you modify the manuscript according to the review recommendations. 

Sincerely,

Aysegul Taylan Ozkan, M.D., Ph.D.,

Academic Editor

Amy Morrison

Section Editor

Reviewer's Responses to Questions

**Key Review Criteria Required for Acceptance?**

**Methods**

-Are the objectives of the study clearly articulated with a clear testable hypothesis stated?

-Is the study design appropriate to address the stated objectives?

-Is the population clearly described and appropriate for the hypothesis being tested?

-Is the sample size sufficient to ensure adequate power to address the hypothesis being tested?

-Were correct statistical analysis used to support conclusions?

-Are there concerns about ethical or regulatory requirements being met?

Reviewer #1: (No Response)

Reviewer #2: Revised manuscript seems convincing and well done.

Reviewer #3: (No Response)

**Results**

-Does the analysis presented match the analysis plan?

-Are the results clearly and completely presented?

-Are the figures (Tables, Images) of sufficient quality for clarity?

Reviewer #1: (No Response)

Reviewer #2: (No Response)

Reviewer #3: (No Response)

**Conclusions**

-Are the conclusions supported by the data presented?

-Are the limitations of analysis clearly described?

-Do the authors discuss how these data can be helpful to advance our understanding of the topic under study?

-Is public health relevance addressed?

Reviewer #1: (No Response)

Reviewer #2: (No Response)

Reviewer #3: (No Response)

**Editorial and Data Presentation Modifications?**

Reviewer #1: Line 34-35: suggest change “lower body weight and increased body temperature” to “lower human body weight and increased human body temperature” to make it clear that it’s not these parameters in the flies that are being referred to.

Line 48 : “leading cause of infectious blindness worldwide.” Please change to “leading infectious cause of blindness worldwide.” The blindness itself is not infectious.

Line 58/59: two “found”s in this sentence

Line 59-61: “The findings suggest that controlling fly populations or preventing fly-eye contact in populations with a high prevalence of trachoma may help control the disease.” I think this is making too much of the data. They’re useful data, but the associations found don’t mean that an apparently relevant intervention will alter prevalence.

Line 88-89: the material in parentheses “(one component of the environmental [E] interventions used to control 89 trachoma)”. This is both awkward from a flow perspective and internally grammatically incorrect: E is environmental improvement, and “environmental interventions” does not really make sense. Suggest delete it.

Line 90: I don’t think it’s correct to say that E interventions are rarely implemented. You could say that specific fly control interventions are rarely implemented, or that E interventions are generally the province of actors outside the trachoma elimination programme, or something along those lines, if you want to. There are many organizations, both governmental and non-governmental, that are doing a lot of work to improve access to WASH in rural communities where trachoma is endemic.

Lines 91 and 100 both refer to “gaps in our knowledge”. Can the authors use a different phrasing for one of these instances?

Line 174: suggest indicate why the 13th household was not included; otherwise the reader is left wondering if this was a resource or administrative decision or if the household just refused to be involved. I don’t think this alters interpretation. It just seems an omission to not provide the rationale.

Line 192-193: “to detect possible inhibition and provide the RPP30 gene internal control”. I think I don’t fully understand this. Do the authors mean to say, “to detect possible inhibition of PCR *using* the RPP30 gene as in internal control”?

Line 269: please change the en-dashes (after each use of “inflammation”) to em-dashes, as called for in the annex to the report on the 4th Global Scientific Meeting on Trachoma

Line 416: suggest change to “person-level” variables

Line 438: I think “bacterium” should be “bacterial” here

Line 438 and 439: I am not sure what the authors mean by “competition” in this context

Line 462-464: “but among infected children contact levels were observed at this higher level into older age groups (until approximately nine years) than in uninfected children” – this does not make sense to me. “Than” indicates a comparison. What is the comparison that is being made?

Reviewer #2: (No Response)

Reviewer #3: (No Response)

**Summary and General Comments**

Reviewer #1: This is a nice set of studies that are generally well described in the paper and which contribute significant data. My biggest comment is that - partly due to the number of different kinds of experiment included in an attempt to generate an overall thesis on Ct transmission – I personally found it hard to piece together the evidence for internal carriage of Ct. The abstract says that “Testing for Ct on field-caught M. sorbens indicated a possibly greater role for internal carriage than that on exterior surfaces.” but as I read the body of the paper my impression had been that the evidence on internal carriage actually came from studies on lab-reared flies, or perhaps comparisons of data on lab-reared and field-caught flies. Some more signposting about the significance of particular experiments and the data that came from them would be helpful in this regard. 

For example: I think in lines 395-401 the authors are describing analyses on dissected parts of lab-reared flies that (a) had been fed solutions containing Ct, then (b) washed so that any external Ct would be removed and (c) homogenized so that internal Ct DNA would be available for PCR. The results then tell me that Ct DNA was present inside the flies. Is that interpretation correct? I’m sure my uncertainty here is related to unfamiliarity with some of the methods, but to help readers like me it might be helpful just to be a bit more explicit in the results and discussion sections.

Reviewer #2: (No Response)

Reviewer #3: (No Response)

PLOS authors have the option to publish the peer review history of their article (what does this mean?). If published, this will include your full peer review and any attached files.

Reviewer #1: No

Reviewer #2: Yes: Hamidreza Hasani

Reviewer #3: No

Figure Files:

Data Requirements:

Reproducibility:

References

---

## [Decision Letter · Decision Letter 2]

7 Jun 2024

Dear Mr Macleod,

We are pleased to inform you that your manuscript 'Field- and laboratory-based studies on correlates of Chlamydia trachomatis transmission by Musca sorbens: determinants of fly-eye contact and investigations into fly carriage of elementary bodies' has been provisionally accepted for publication in PLOS Neglected Tropical Diseases.

Best regards,

Aysegul Taylan Ozkan, M.D., Ph.D.,

Academic Editor

Amy Morrison

Section Editor

Reviewer's Responses to Questions

**Key Review Criteria Required for Acceptance?**

**Methods**

-Are the objectives of the study clearly articulated with a clear testable hypothesis stated?

-Is the study design appropriate to address the stated objectives?

-Is the population clearly described and appropriate for the hypothesis being tested?

-Is the sample size sufficient to ensure adequate power to address the hypothesis being tested?

-Were correct statistical analysis used to support conclusions?

-Are there concerns about ethical or regulatory requirements being met?

Reviewer #1: Yes

Yes

Yes

Yes

Yes

No

**Results**

-Does the analysis presented match the analysis plan?

-Are the results clearly and completely presented?

-Are the figures (Tables, Images) of sufficient quality for clarity?

Reviewer #1: Yes

Yes

Yes

**Conclusions**

-Are the conclusions supported by the data presented?

-Are the limitations of analysis clearly described?

-Do the authors discuss how these data can be helpful to advance our understanding of the topic under study?

-Is public health relevance addressed?

Reviewer #1: Yes

Yes

Yes

Yes

**Editorial and Data Presentation Modifications?**

Reviewer #1: Lines 242 and 244: “puparium” sounds singular to me. Is the plural “puparia”?

273: please use em-dashes, not hyphens in “trachomatous inflammation-follicular” and “trachomatous inflammation-intense

299: please add “years” after “aged 1 to 50”

466: “principle” here should be “principal”

**Summary and General Comments**

Reviewer #1: Now very clear. This will be an excellent addition to the trachoma literature.

PLOS authors have the option to publish the peer review history of their article (what does this mean?). If published, this will include your full peer review and any attached files.

Reviewer #1: No

---

## [Editor Report · Acceptance letter]

27 Jun 2024

Dear Mr Macleod,

We are delighted to inform you that your manuscript, "Field- and laboratory-based studies on correlates of Chlamydia trachomatis transmission by Musca sorbens: determinants of fly-eye contact and investigations into fly carriage of elementary bodies," has been formally accepted for publication in PLOS Neglected Tropical Diseases.

Best regards,

Shaden Kamhawi

co-Editor-in-Chief

Paul Brindley

co-Editor-in-Chief
